# Association of Baseline Lipopolysaccharide-Binding Protein with Expanded Disability Status Score Dynamics in Patients with Relapsing–Remitting Multiple Sclerosis: A Pilot Study

**DOI:** 10.3390/ijms26010298

**Published:** 2024-12-31

**Authors:** Anda Vilmane, Oksana Kolesova, Zaiga Nora-Krukle, Aleksandrs Kolesovs, Daina Pastare, Liga Jaunozolina, Linda Kande, Jelena Egle, Daniela Kromane, Madara Micule, Sintija Liepina, Estere Zeltina, Sabine Gravelsina, Santa Rasa-Dzelzkaleja, Ludmila Viksna, Guntis Karelis

**Affiliations:** 1Institute of Microbiology and Virology, Rīga Stradiņš University Research Center, LV-1067 Riga, Latvia; 2Department of Infectology, Rīga Stradiņš University, LV-1007 Riga, Latvia; 3Department of Neurology and Neurosurgery, Riga East University Hospital, LV-1038 Riga, Latvia; 4Department of Neurology and Neurosurgery, Rīga Stradiņš University, LV-1002 Riga, Latvia; 5Center of Radiology, Riga East University Hospital, LV-1038 Riga, Latvia; 6Department of Radiology, Rīga Stradiņš University, LV-1079 Riga, Latvia; 7Faculty of Medicine, Rīga Stradiņš University, LV-1007 Riga, Latvia; 8Department of Residency, Rīga Stradiņš University, LV-1007 Riga, Latvia; 9Riga East University Hospital, LV-1038 Riga, Latvia

**Keywords:** relapsing–remitting multiple sclerosis, lipopolysaccharide-binding protein, neuroinflammation, EDSS

## Abstract

Forecasting the progression of the disease in the early inflammatory stage of the most prevalent type of multiple sclerosis (MS), referred to as relapsing–remitting multiple sclerosis (RRMS), is essential for making prompt treatment modifications, aimed to reduce clinical relapses and disability. In total, 58 patients with RRMS, having an Expanded Disability Status Scale (EDSS) score less than 4, were included in this study. Baseline magnetic resonance imaging (MRI) was performed, and brain and spinal cord lesions were evaluated. The disability of the patients was evaluated using EDSS at baseline and follow-up; enzyme-linked immunosorbent assays (ELISAs) were also used to determine the level of blood-based inflammation markers in plasma at baseline. The main results demonstrated that the baseline level of LBP was correlated with an increase in EDSS in a short (8–10 months) follow-up period. Furthermore, the prognostic significance of LBP was only observed in patients who received disease-modifying treatment (DMT) before the study. Our results suggest that the baseline level of LBP may be among the predictors of disability progression in RRMS over short follow-up periods, particularly in those receiving treatment. It highlights the effect of endotoxins in the pathogenesis of RRMS.

## 1. Introduction

Multiple sclerosis (MS) is a chronic autoimmune inflammatory condition of the central nervous system (CNS) marked by the destruction of myelin and subsequent damage to axons, leading to a deterioration in both motor and sensory abilities. It involves activated T cells, and there is increasing evidence that B cells and components of the innate immune system also play a significant role in the disease [1]. Relapsing–remitting multiple sclerosis (RRMS) is the most common form of MS, characterized by episodes of neurological symptoms (relapses) followed by periods of recovery (remissions). New symptoms may appear during a relapse, or existing symptoms may worsen. These episodes can vary in duration and severity, potentially lasting days to weeks or even longer. Common symptoms during relapses can include fatigue, numbness, weakness, vision problems, difficulty walking, and cognitive changes. After a relapse, individuals typically experience remission with some level of recovery where symptoms stabilize or improve. In many cases, individuals return to their pre-relapse functional status. However, complete recovery is not always guaranteed, and some residual symptoms may persist. Over time, RRMS can progress to secondary progressive MS, where individuals experience a gradual worsening of symptoms and disability, with or without occasional relapses [2].

The Expanded Disability Scale (EDSS) is a clinical rating system used to assess the disability level and functional status of individuals with MS [3]. It ranges from 0 to 10, with lower scores indicating no disability and higher scores increasing degrees of disability. The scale evaluates various neurological functions, including mobility, visual abilities, coordination, and sensory functions, providing a comprehensive view of a patient’s condition over time. Clinicians use the EDSS to track disease progression and guide treatment plans. However, the disease can progress without symptoms, which highlights the need for identifying new biomarkers for subclinical disease progression [4].

Predicting the disease course during the early inflammatory phase of RRMS is crucial for timely treatment adjustments, minimizing clinical relapses and disability, and achieving improved long-term outcomes. Based on previous studies, neuroinflammation can be assessed not only by the presence of gadolinium-enhancing lesions in magnetic resonance imaging (MRI) but also by different markers in the cerebrospinal fluid (CSF) [5,6,7,8]. However, the use of CSF for monitoring the disease is limited due to the invasiveness of the procedure, and it is usually performed for the diagnosis of MS [9].

Blood-based markers offer a practical approach for detecting and monitoring various stages of MS, potentially providing valuable insights into disease activity over time. Previous studies have highlighted their associations with key pathological processes such as neuroinflammation and neurodegeneration. For instance, routine clinical tests have explored the neutrophil-to-lymphocyte ratio (NLR) as a marker of systemic inflammation, while neurofilament light chain (NfL) has been identified as a reliable marker of axonal damage and neurodegeneration, showing promise for tracking disease progression [5,10]. Markers of immune activation showed potential associations with neuroinflammation, including the soluble B-cell maturation antigen (sBCMA) and soluble CD14 (sCD14), which is produced by activated macrophages [8,11,12]. Lipopolysaccharide (LPS)-binding protein (LBP) is an acute-phase protein, facilitating its recognition by immune cells, that has been discussed in the context of disrupted intestinal barrier homeostasis and the endotoxin hypothesis of neurodegeneration [13,14,15,16,17]. In conditions such as MS and other neurodegenerative disorders, the blood–brain barrier (BBB)’s integrity can be compromised. This compromise can allow neurotoxic substances, including LPS, to enter the brain, amplifying neuroinflammatory responses [18].

The aim of this pilot study was to evaluate the baseline levels of blood-based markers (BCMA, sCD14, IFN-γ, and LBP) as possible predictors of changes in disability in the early phase of RRMS. For the assessment of disability, an EDSS score was used.

## 2. Results

### 2.1. Baseline Evaluation

Patients with RRMS (*n* = 58) in this cohort were aged 18 to 67 years with a mean age of 38 ± 12 years; the healthy control (HC) group (*n* = 56) was aged 29 to 61 years with a mean age of 40 ± 10 years (U = 1425.0; *p* = 0.259). In the RRMS group, 43 patients (74%) were female, while in the HC group, 32 (57%) were female (χ^2^ = 3.65; *p* = 0.056). Twenty-seven patients (47%) were under disease-modifying treatment (DMT), which was started after the confirmation of the diagnosis. Table 1 presents the demographic and clinical characteristics of RRMS patients according to the presence of previous treatment before the study.

Differences between subgroups have been observed for three parameters: the median disease duration since the diagnosis, the median disease duration since the first symptoms, and the presence of optic nerve demyelination. Patients under DMT at the baseline assessment had a longer disease duration and more frequent optic nerve demyelination than patients with newly diagnosed RRMS. There were no differences in demographics, body mass index (BMI), and parameters associated with neurological disability, disease activity such as EDSS and number of Gd+ lesions in the MRI, and inflammatory markers such as NLR, BCMA, sCD14, IFN-γ, and LBP levels at baseline. Therefore, the comparison of inflammatory markers between patients and the control group was performed involving the joint RRMS group (*n* = 58) and the HC group (*n* = 56).

At baseline, patients with RRMS had a higher level of sCD14 (*p* = 0.033) and a higher level of IFN-γ (*p* < 0.001) than the HC group. In contrast, patients with RRMS had a lower level of BCMA than the HC group (*p* < 0.001). There were no differences in the LBP level (Table 2).

### 2.2. Follow-Up Evaluation

A follow-up visit was carried out for 50 (86.2%) RRMS patients. Within 8 to 10 months, EDSS changed in 29 out of 50 cases (58%). It decreased for eight patients (16%) and increased for thirteen (26%). However, only one patient had a relapse during the follow-up period. To explore parameters for predicting the EDSS change for 8–10 months, we performed Spearman’s correlation analysis between EDSS change and demographics, disease duration, number of active lesions, presence of brain and spinal cord atrophy, previous treatment, and parameters at baseline.

Correlation analysis (Table 3) showed that EDSS change links to four parameters: (1) LBP in plasma at baseline (r_s_ = 0.30, *p* < 0.05); (2) disease duration since the first symptoms (r_s_ = 0.29, *p* < 0.05); (3) previous DMT (r_s_ = 0.39. *p* < 0.01), indicating that EDSS was increased during follow-up in patients with treatment at baseline; and (4) the presence of optic nerve demyelination (r_s_ = 0.30, *p* < 0.05).

In the next step, we applied linear regression to explore a predictive model for EDSS change by parameters showing significant correlation with it (LBP; disease duration since first symptoms; previous DMT; and the presence of optic nerve demyelination) as predictors. The analysis showed that EDSS change can be significantly predicted by DMT received before the baseline and LBP level in plasma at baseline (Table 4). The EDSS change was not predicted by the disease duration since the first symptoms or the presence of the optic nerve demyelination. The normal distribution of residuals supported the application of linear regression. Figure 1a presents the regression of EDSS change on the LBP level in all patients.

To specify effects in groups with and without previous treatment, we compared results between RRMS patients who previously received DMT and RRMS patients who did not receive DMT at the baseline. Table 5 includes regression models for both groups of patients. The analysis showed that the LBP level predicts EDSS change only in the previously treated patient subgroup, as shown in Figure 1b. In these patients, an increase in EDSS in 8–10 months of follow-up was associated with a higher level of plasma LBP at baseline.

## 3. Discussion

The main results demonstrated that the baseline level of LBP can be among the predictors of change in EDSS in a short (8–10 months) follow-up period. However, the prognostic significance of LBP was observed only in patients who received DMT before the study. On the one hand, the absence of differences in EDSS and inflammatory markers at baseline between previously treated and untreated patients in our study indicated a protective effect of DMT on inflammation in patients because previously treated patients had a longer MS duration since the confirmation of the diagnosis and since the first symptoms and had more frequent demyelination of the optic nerve than in patients with newly diagnosed MS cases. On the other hand, the observed tendency of increasing disability in previously treated patients with a higher level of LBP may point to residual inflammation not reduced by DMT.

LBP is an indirect marker of endotoxemia, predominantly released by hepatocytes triggered by LPS—the molecules of the outer membrane of Gram-negative bacteria. After binding with LPS, the complex of LBP and LPS is eliminated from circulation by the CD14 receptor on Kupffer cells [19]. Therefore, sCD14 is also an indirect marker of endotoxemia. Based on our results, patients with RRMS had a higher level of sCD14 than the HC group, indicating potentially higher levels of endotoxins in patients. According to Brown 2019, endotoxins can be chronically elevated in the blood due to altered integrity of the mucosa of the intestine, oral cavity, and other tissues. It increases the permeability of the blood–brain barrier (BBB), leading to the crossing of autoreactive T and B lymphocytes into the central nervous system. Endotoxins can also cross the BBB and trigger microglial cells, increase neuroinflammation, and, therefore, directly contribute to the damage of myelin sheath by the immune cells [15]. The observed association between a higher baseline level of LBP and worsening of the disease course assessed by EDSS is in line with the endotoxin hypothesis contributing to neurodegeneration, despite no clinically documented relapse during the follow-up. However, for more precious assessment, the worsening of the disease course should be confirmed within the defined time interval (for example, in six months), which was not performed in our study [20]. Therefore, a more detailed analysis of disease dynamics at different stages remains the question for further longitudinal research.

Our findings indirectly support the opinion that correcting and reducing endotoxin levels combined with disease-modifying treatment may delay disability progression [13]. Given the role of LBP in modulating immune responses and its connection to BBB integrity, targeting LBP or its pathways may offer therapeutic potential. Strategies that modulate the interaction between LBP and the innate immune response could help maintain BBB integrity and reduce neuroinflammation, possibly providing protective effects in neurodegenerative diseases. The studies conducted in animal models of MS have shown that treatment with enzymes such as alkaline phosphatase can neutralize the effects of endotoxins and reduce inflammatory responses, which may reduce the severity of the disease [21].

At the same time, our study has significant limitations. The sample size is small despite including nearly the entire cohort of patients in our country within the timeframe of a pilot study. The observed trend demonstrated variability in smaller subgroups, and numerically larger populations are needed to assess the robustness of findings. Specifically, further investigation is required to validate the interaction between chronic endotoxemia and neurological disability and validate the applicability of these findings for therapeutic interventions.

In summary, the study suggests that the level of LBP may be among the predictors of disability progression in RRMS over short follow-up periods, particularly in those receiving treatment. It highlights the possible effect of endotoxins in the pathogenesis of RRMS and neurological disability despite the use of DMT and the absence of new relapses during follow-up. However, further research is needed to better understand the mechanisms through which endotoxins affect disease progression and to explore potential interventions that could mitigate their impact on patients with RRMS.

## 4. Materials and Methods

### 4.1. Procedure and Participants

The pilot study was conducted from February 2022 to September 2024 in the tertiary hospital in Riga, Latvia. The central role of a selected hospital at the country level determined that the study involved almost the entire cohort of new and current patients who provided their consent within the timeframe of a pilot study and fitted the inclusion criteria.

Inclusion criteria for patients were adult age and newly or previously diagnosed RRMS with EDSS score < 4. Pregnancy, oncological, and other autoimmune diseases were exclusion criteria for the study. All participants signed an informed consent form. The study was approved by the Rīga Stradiņš University Research Ethics Committee (protocol code: 2-PĒK-4/35/2023).

In the prospective study, 58 patients were included; 27 (46.6%) of them had disease-modifying treatment (DMT) before the beginning of the study. Other patients (previously untreated patients) started treatment after the blood samples were obtained. A follow-up visit was carried out for 50 (86.2%) patients 8 to 10 months later. In addition, plasma samples from age- and sex-matched healthy individuals were included to serve as controls (HC group). None of them (*n* = 56) had any history of neurological diseases.

### 4.2. Clinical Evaluation and MRI

A certified neurologist conducted the clinical evaluation of RRMS patients and assessed disability using the Expanded Disability Status Scale (EDSS) at baseline and after 8 to 10 months. Baseline magnetic resonance imaging (MRI) was performed, and brain and spinal cord lesions were evaluated according to practical guidelines by a certified neuroradiologist [7]. Additionally, peripheral blood samples for detection of markers of inflammation were obtained after clinical evaluation at baseline and at follow-up.

### 4.3. Enzyme-Linked Immunosorbent Assay (ELISA)

Plasma samples were isolated by centrifugation at 2000× *g* at 4 °C for 10 min, aliquoted, and frozen at −80 °C until analysis. Plasma BCMA, sCD14, LBP, and IFN-γ were measured by ELISA according to the manufacturer’s instructions: Human TNFRSF17/BCMA ELISA Kit PicoKine^®^, cat. No EK0661 (Boster Biological Technology, Pleasanton, CA, USA); BioOcean^®^ Human sCD14 ELISA Kit, cat. No EK1101 (Bio-Ocean LLC, Arden Hills, MN, USA); Enzyme Immunoassay for Quantification of free human LBP, cat. No 044 (Chromatec GmbH, Greifswald, Germany); Human IFNg ELISA Kit, cat. No ELK1036 (ELK (Wuhan) Biotechnology Co., Ltd., Wuhan, China).

### 4.4. Statistics

The Shapiro–Wilk test indicated significant deviance from the normal distribution in most of the variables in the clinical and control groups. Therefore, we applied the Mann–Whitney test to compare subgroups. Spearman’s rank correlation coefficient and a biserial correlation coefficient were used to assess relationships between EDSS and continuous and binary variables, respectively. The distribution of categorical variables in subgroups was assessed with the chi-square test. The predictive model for EDSS change involved linear regression. We have considered the a priori sample size using an online calculator for multiple regression [22], accounting for an alpha level of 0.05, a power of 0.80, up to nine predictors, and an effect size of 0.33, corresponding to a medium coefficient of determination of 0.25 [23]. It resulted in a minimal sample size of 57 participants, confirming that the available number of participants is acceptable for the exploratory testing of the model. The actual number of predictors was four, corresponding to the sample size of 41 participants. All calculations were performed using IBM SPSS Statistics for Windows 29 (Armonk, NY, USA) at a statistical significance threshold of *p* < 0.05.

## Figures and Tables

**Figure 1 ijms-26-00298-f001:**
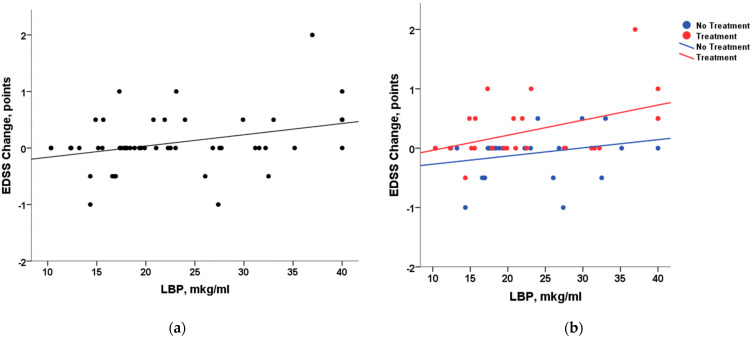
Linear regression of EDSS changes after 8–10 months on LBP level: (**a**) in all patients (*n* = 50); (**b**) in patients with (*n* = 24) and without (*n* = 26) previous treatment.

**Table 1 ijms-26-00298-t001:** Demographic and clinical characteristics of RRMS patients at baseline.

Parameters	*n*	Previously Treated Patients	*n*	Previously Untreated Patients	*p*-Value	Total
Mean age; years (SD)	27	38 (10)	31	37 (13)	0.478	38 (12)
Females; *n* (%)	27	20 (74%)	31	23 (74%)	0.992	43 (74%)
BMI; kg/m^2^ (IQR)	26	26.0 (24.0–28.8)	28	25.3 (21.2–29.2)	0.451	25.8 (22.2–29.1)
Smokers; *n* (%)	27	6 (22%)	28	12 (43%)	0.103	18 (31%)
Median EDSS; score (IQR)	26	2.0 (1.5–2.1)	31	2.0 (1.5–2.5)	0.663	2 (1.5–2.5)
Median disease duration after diagnosis; months (IQR)	25	38 (15–74)	25	1 (1–2)	<0.001	10 (1–42)
Median disease duration since first symptoms; months (IQR)	27	72 (33–96)	28	11 (1–37)	<0.001	36 (7–86)
Median number of relapses since diagnosis; *n* (IQR)	25	1 (1–3)	26	1 (1–2)	0.754	1 (1–2)
Median T_2_ lesion count; *n* (IQR)						
Periventricular	27	10 (4–20)	31	10 (3–20)	0.796	10 (4–20)
Juxtacortical		6 (2–12)		4 (1–15)	0.547	5 (2–13)
Infratentorial		3 (1–6)		3 (1–7)	0.570	3 (1–6)
Cervical spinal cord		2 (0–4)		2 (0–3)	0.336	2 (0–4)
Thoracic spinal cord		1 (0–3)		1 (0–3)	0.662	1 (0–3)
Conus medullaris		0 (0–0)		0 (0–0)	0.735	0–2
Patients with Gd+ lesions; *n* (%)	27	5 (19%)	30	12 (40%)	0.077	16 (28%)
Presence of optic nerve demyelination; *n* (%)	27	21 (78%)	28	10 (36%)	0.002	19 (61%)
Presence of optic nerve atrophy; *n* (%)	27	4 (15%)	28	3 (11%)	0.648	8 (14%)
Presence of brain atrophy; *n* (%)	25	4 (16%)	28	8 (29%)	0.275	12 (22%)
Presence of spinal cord atrophy; *n* (%)	25	1 (4%)	28	2 (7%)	0.621	3 (6%)

BMI—body mass index; DMT—disease-modifying treatment; Gd+ lesions—gadolinium-enhancing lesions.

**Table 2 ijms-26-00298-t002:** Plasma inflammatory markers in RRMS patients at baseline and in HC group.

Parameters	Patients with RRMS at Baseline (*n* = 58)	Healthy Control (*n* = 56)	U-Value	*p*-Value
Median BCMA (IQR), ng/mL	308.2 (260.4–338.4)	403.7 (370.7–433.6)	518.0	<0.001
Median sCD14 (IQR), ng/mL *	1729 (1388–1975)	1490 (1251–1805)	1225.0	0.033
Median IFN-γ (IQR), pg/mL *	148.7 (88.1–236.6)	89.9 (30.1–107.7)	773.0	<0.001
Median LBP (IQR), ng/mL	20,325 (16,360–28,250)	21,445 (16,530–29,763)	1551.0	0.679

U—Mann–Whitney test; *—57 out of 58 RRMS patients analyzed.

**Table 3 ijms-26-00298-t003:** Spearman’s correlation coefficients for EDSS change in 8–10 months and demographic and clinical indicators.

Parameters	Number of RRMS Patients Analyzed (*n*)	r_s_ (With EDSS Change)	*p*-Value
Age	50	−0.01	0.966
Sex ^a^	50	−0.02	0.914
BMI	50	−0.01	0.973
DMT at baseline ^a^	50	0.39	0.005
Disease duration since 1-st symptoms	50	0.29	0.046
Number of relapses	45	0.13	0.387
Time since the last relapse ^b^	24	0.30	0.160
BCMA, baseline	50	0.08	0.561
sCD14, baseline	46	0.10	0.511
IFN-γ, baseline	46	0.00	0.998
LBP, baseline	50	0.30	0.032
NLR, baseline	50	−0.06	0.658
EDSS, baseline	50	−0.06	0.658
Gd+ lesion count, at baseline	50	−0.18	0.223
Optic nerve demyelination ^a^	48	0.30	0.036
Brain atrophy ^a^	46	−0.27	0.074
Spinal cord atrophy ^a^	46	−0.23	0.126

^a^ Biserial correlation coefficient was applied for binary variables; ^b^ previously treated patients.

**Table 4 ijms-26-00298-t004:** Linear regression for predicting EDSS change in 8–10 months (*n* = 46).

Variables	B	S.E.	β	t	*p*
**Patients with previous DMT**	0.357	0.151	0.352	2.37	0.022
**LBP**	0.019	0.008	0.316	2.30	0.026
**Time from first symptoms**	0.000	0.001	−0.015	−0.106	0.916
**Optic nerve demyelination**	0.115	0.159	0.111	0.720	0.476
**Constant**	−0.575	0.209		−2.75	0.009

Model characteristics: F (4; 41) = 4.11, *p* = 0.007. R^2^ = 0.29. B—non-standardized regression coefficient; S.E.—standard error; β—standardized regression coefficient; *p*—significance level.

**Table 5 ijms-26-00298-t005:** Linear regressions for predicting EDSS change in 8–10 months by LBP in previously treated (*n* = 24) and untreated (*n* = 26) RRMS patients.

Variables	B	S.E.	β	t	*p*
**Previously treated patients**					
**LBP**	0.025	0.012	0.414	2.13	0.044
**Constant**	−0.290	0.291		−1.00	0.330
Model characteristics: F (1; 22) = 4.55, *p* = 0.044, R^2^ = 0.17					
**Previously untreated patients**					
**LBP**	0.014	0.009	0.286	1.46	0.156
**Constant**	−0.408	0.226		−1.80	0.084
Model characteristics: F (1; 24) = 2.14, *p* = 0.156, R^2^ = 0.08					

B—non-standardized regression coefficient; S.E.—standard error; β—standardized regression coefficient; *p*—significance level.

## Data Availability

The original data presented in the study are openly available in Rīga Stradiņš University Institutional Repository Dataverse at https://doi.org/10.48510/FK2/UWN5EV (accessed on 9 December 2024).

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
