# Peer review of "Association of Baseline Lipopolysaccharide-Binding Protein with Expanded Disability Status Score Dynamics in Patients with Relapsing–Remitting Multiple Sclerosis: A Pilot Study"

_ijms, 2024, doi:10.3390/ijms26010298_

Round 1
Reviewer 1 Report
Comments and Suggestions for Authors
The manuscript is well organized and written, giving to the potential readers all information able to understand the purpose of the work. The design of the study is clear and the methods are described in details. The results summarizes all obtained findings and are well contextualized in the discussion section.
I suggest to the authors to specify in the statistical methods the way in which they analyzed the distribution of the data (i.e., Shapiro-Wilk) and if they have a parametric or non-parametric distribution, justifying in this way the reason why they used Mann-Whitney test and Spearman's correlation to analyze their data.
English language is good, leading no problem for the comprehension of the whole text.
Author Response
Dear Reviewer,
Thank you very much for your thoughtful and positive feedback on our article.
We have added the description of the distribution-related test.
Reviewer 2 Report
Comments and Suggestions for Authors
This is a study on blood biomarkers in multiple sclerosis. The study investigated the usefulness of biomarkers such as LBP as predictors of neurological damage in patients with early relapsing-remitting MS.
Although the cohort is consisted of relatively large number of patients (n=58), the patients were heterogeneous in many ways and these differences are confounding factors. Subgroup analyses should therefore be carefully performed.
1. more detailed disease-stage analysis
The duration of disease ranges from 1 to 42 months, and the time since last relapse is expected to vary considerably (the time since last relapse should also be described in the table). It is expected that the immunopathology of each of these groups will be different, e.g. immediately after relapse, immediately after acute treatment such as steroid pulses, less than 1 year after relapse, stable for more than 2 years after relapse, etc., so it would be inappropriate to analyse them in a combined manner.
2. the influence of therapeutic drugs
As the patients in this study were not randomly allocated with or without DMD treatment, the use of DMD should have been chosen according to the condition (circumstances) of each patient, so the demography of the two groups before treatment should be considered different. For example, patients being followed up without DMD may have a mild or asymptomatic condition; what would be the difference in the demography of the two groups at the time of DMD introduction?
3. Longitudinal analysis
In view of the above issues, an analysis examining changes in markers at different stages of disease in a single patient would be desirable. Why not mention this in the discussion?
4. details of EDSS changes
EDSS is a combination of various symptoms and the same EDSS change can vary in content and severity. Readers may be interested in the specific changes observed.
Author Response
Dear Reviewer,
Thank you for your comprehensive review and for taking the time to provide us with your valuable feedback on our manuscript. While we recognize that there are areas for improvement, we greatly appreciate your constructive suggestions, which we believe will enhance the quality of our work.
Comments 1: more detailed disease-stage analysis
The duration of disease ranges from 1 to 42 months, and the time since last relapse is expected to vary considerably (the time since last relapse should also be described in the table). It is expected that the immunopathology of each of these groups will be different, e.g. immediately after relapse, immediately after acute treatment such as steroid pulses, less than 1 year after relapse, stable for more than 2 years after relapse, etc., so it would be inappropriate to analyse them in a combined manner.
Response 1: We have changed Table 1 by adding a more detailed description of the patients’ subgroups (treated and untreated at the beginning of the study) and a comparison of groups.
Comments 2: the influence of therapeutic drugs
As the patients in this study were not randomly allocated with or without DMD treatment, the use of DMD should have been chosen according to the condition (circumstances) of each patient, so the demography of the two groups before treatment should be considered different. For example, patients being followed up without DMD may have a mild or asymptomatic condition; what would be the difference in the demography of the two groups at the time of DMD introduction?
Response 2: We have added demography. For the treated group, DMT has been started since the diagnosis confirmation. Untreated patients had no DMT at the beginning of the study (baseline) due to the need for diagnosis specification and identifying indications for treatment. Therefore, DMT was started for previously untreated patients after obtaining the samples.
Comments 3: Longitudinal analysis
In view of the above issues, an analysis examining changes in markers at different stages of disease in a single patient would be desirable. Why not mention this in the discussion?
Response 3: The information examining changes in markers at different stages of disease were not included in the evaluation because of limitations for the pilot project. Therefore, the prospective assessment has included only EDSS change after 8-10 months since the baseline evaluation.
Comments 4: details of EDSS changes
EDSS is a combination of various symptoms and the same EDSS change can vary in content and severity. Readers may be interested in the specific changes observed.
Response 4: Initial evaluation of the functional systems in EDSS did not show an association with plasma markers at baseline, emphasizing the value of a composite value. Therefore, we did not include these subscales in the assessment.
Reviewer 3 Report
Comments and Suggestions for Authors
While the study addresses a clinically relevant topic, some weaknesses limit its contribution to the field.
The study design does not adequately justify its small sample size or address its implications for statistical power. A "pilot study" label does not excuse the lack of a power calculation or clear rationale for the chosen cohort size.
The reported associations between LBP and EDSS change are weak, with small effect sizes and low R2 values in regression models (e.g., R²2= 0.28). These findings are insufficiently robust to support the claim that LBP is a predictive biomarker.
The study extrapolates clinical relevance from statistically modest results. For example, the observed association between LBP and EDSS progression is limited to treated patients and does not replicate in untreated ones, suggesting potential confounding rather than a clear biological effect.
Grammatical errors, typographical inconsistencies, and unclear figure legends.
Comments on the Quality of English LanguageThe English should be improved
Author Response
Dear Reviewer,
Thank you for your review and for sharing your perspectives on our manuscript. We appreciate the constructive criticism and suggestions you have provided, as they are essential for reinforcing the rigor and clarity of our research.
Comments 1: The study design does not adequately justify its small sample size or address its implications for statistical power. A "pilot study" label does not excuse the lack of a power calculation or clear rationale for the chosen cohort size.
Response 1: We have added consideration for the sample size.
Comments 2: The reported associations between LBP and EDSS change are weak, with small effect sizes and low R2 values in regression models (e.g., R²2= 0.28). These findings are insufficiently robust to support the claim that LBP is a predictive biomarker.
Response 2: We have changed the claim regarding the biomarker to a hypothetical one.
Comments 3: The study extrapolates clinical relevance from statistically modest results. For example, the observed association between LBP and EDSS progression is limited to treated patients and does not replicate in untreated ones, suggesting potential confounding rather than a clear biological effect.
Response 3: We agree to criticism. The conclusions are specified.
Comments 4: Grammatical errors, typographical inconsistencies, and unclear figure legends.
Response 4: We have improved the text and legends throughout the manuscript.
Round 2
Reviewer 2 Report
Comments and Suggestions for Authors
significantly improved.
Author Response
Dear Reviewer,
Thank you for your positive feedback.
Reviewer 3 Report
Comments and Suggestions for Authors
The authors have made revisions to address the feedback; however, significant issues persist. The small sample size still lacks a robust power calculation, undermining the reliability and generalizability of the findings. Additionally, the weak associations and low effect sizes, coupled with limited replication across groups, fail to support the study’s claims, even in their revised hypothetical framing.
The conclusions continue to overstate clinical relevance, with confounding factors likely driving the observed associations. These modest results do not provide sufficient evidence for the proposed biomarker’s predictive utility or broader implications.
Author Response
Dear Reviewer,
Thank you for your comments. On the one hand, we recognize weaknesses of our study. We have extended the description of the sample in Methods and formulated limitations in Discussion. On the other hand, some criticism refers to unchangeable issues. We have included almost the full cohort of patients in our country within the timeframe of a pilot study. The identified trend was tested for its stability in subgroups and its variability was identified. Therefore, we can hypothesize rather than make a conclusion. We have made our conclusions more cautious and hypothetical. We hope these modifications align better with your expectations and enhance the clarity of our conclusions. Thank you once again for your valuable input.
Round 3
Reviewer 3 Report
Comments and Suggestions for Authors
While I appreciate the modifications the authos have made. The study continues to suffer from limitations inherent in its small sample size, which is not sufficiently justified by a power calculation. The weak statistical associations, particularly the limited replication of results in untreated patients, raise doubts about the robustness of the findings. Furthermore, while the authors have revised their claims regarding LBP as a predictive biomarker, the lack of a solid statistical foundation leaves the conclusions tenuous and speculative at best. Given these ongoing issues, I believe the manuscript should not be considered by the jouranl.